# The Immune System Response to *Porphyromonas gingivalis* in Neurological Diseases

**DOI:** 10.3390/microorganisms11102555

**Published:** 2023-10-13

**Authors:** Raffaella Franciotti, Pamela Pignatelli, Domenica Lucia D’Antonio, Rosa Mancinelli, Stefania Fulle, Matteo Alessandro De Rosa, Valentina Puca, Adriano Piattelli, Astrid Maria Thomas, Marco Onofrj, Stefano Luca Sensi, Maria Cristina Curia

**Affiliations:** 1Department of Neuroscience, Imaging and Clinical Science, G. d’Annunzio University of Chieti-Pescara, 66100 Chieti, Italy; rosa.mancinelli@unich.it (R.M.); stefania.fulle@unich.it (S.F.); matteo.derosa1994@gmail.com (M.A.D.R.); athomas@unich.it (A.M.T.); onofrj@unich.it (M.O.); stefano.sensi@unich.it (S.L.S.); 2COMDINAV DUE, Nave Cavour, Italian Navy, Stazione Navale Mar Grande, Viale Jonio, 74122 Taranto, Italy; pamela.pignatelli@marina.difesa.it; 3Department of Medical, Oral and Biotechnological Sciences, G. d’Annunzio University of Chieti-Pescara, 66100 Chieti, Italy; domenica.dantonio@unich.it (D.L.D.); mc.curia@unich.it (M.C.C.); 4Fondazione Villaserena per la Ricerca, 65013 Città Sant’Angelo, Pescara, Italy; 5Center for Advanced Studies and Technology (CAST), G. d’Annunzio University of Chieti-Pescara, 66100 Chieti, Italy; 6Department of Pharmacy, G. d’Annunzio University of Chieti-Pescara, 66100 Chieti, Italy; valentina.puca@unich.it; 7School of Dentistry, Saint Camillus International University for Health Sciences, 00131 Rome, Italy; apiattelli51@gmail.com; 8Facultad de Medicina, UCAM Universidad Católica San Antonio de Murcia, Guadalupe, 30107 Murcia, Spain; 9Institute for Advanced Biomedical Technologies, G. d’Annunzio University of Chieti-Pescara, 66100 Chieti, Italy

**Keywords:** *Porphyromonas gingivalis*, dysbiosis, oral cavity, antibodies, neurological and neurodegenerative diseases, immune system, immunosorbent, acute condition

## Abstract

Previous studies have reported an association between oral microbial dysbiosis and the development and progression of pathologies in the central nervous system. *Porphyromonas gingivalis* (*Pg*), the keystone pathogen of the oral cavity, can induce a systemic antibody response measured in patients’ sera using enzyme-linked immunosorbent assays. The present case–control study quantified the immune system’s response to *Pg* abundance in the oral cavities of patients affected by different central nervous system pathologies. The study cohort included 87 participants: 23 healthy controls (HC), 17 patients with an acute neurological condition (N-AC), 19 patients with a chronic neurological condition (N-CH), and 28 patients with neurodegenerative disease (N-DEG). The results showed that the *Pg* abundance in the oral cavity was higher in the N-DEG patients than in the HC (*p* = 0.0001) and N-AC patients (*p* = 0.01). In addition, the *Pg* abundance was higher in the N-CH patients than the HCs (*p* = 0.005). Only the N-CH patients had more serum anti-*Pg* antibodies than the HC (*p* = 0.012). The inadequate response of the immune system of the N-DEG group in producing anti-*Pg* antibodies was also clearly indicated by an analysis of the ratio between the anti-*Pg* antibodies quantity and the *Pg* abundance. Indeed, this ratio was significantly lower between the N-DEG group than all other groups (*p* = 0.0001, *p* = 0.002, and *p* = 0.03 for HC, N-AC, and N-CH, respectively). The immune system’s response to *Pg* abundance in the oral cavity showed a stepwise model: the response diminished progressively from the patients affected with an acute condition to the patients suffering from chronic nervous system disorders and finally to the patients affected by neurodegenerative diseases.

## 1. Introduction

The oral cavity is characterized by distinctive features, making it an ideal habitat for a collection of microorganisms [1]. The mouth harbors more than 700 bacterial species or phylotypes, which are site- and subject-specific [2].

*Porphyromonas gingivalis* (*Pg*) is a Gram-negative, nonmotile short rod, and an anaerobic bacterium, considered to be the keystone pathogen of the oral cavity because it is implicated in the biofilm formation of bacterial plaques. With other oral microbes, it forms a biofilm that alters tight epithelial junctions and promotes inflammation by mucosal immune cells. *Pg* can modulate the host immune response, at first increasing nutrient availability and biofilm growth, and subsequently facilitating bacterial resistance by destroying complement factors [3]. *Pg* produces many virulence factors, such as proteases, endotoxins, organic acids, and key enzymes, including the proteases known as gingipains [4]. Gingipains adhere to the epithelium, affecting its permeability and upregulating inflammation. The infiltration of immune cells—macrophages, neutrophils, and lymphocytes—leads to an altered antigen presentation, which overturns the equilibrium between the host and the microorganisms, creating dysbiosis [5]. *Pg* inhibits the conversion of M2 macrophages by redirecting them to the M1 inflammatory subtype, worsening the inflammatory environment. M1 may exhibit cytotoxic and M2 neuroprotective effects [6]. Chronic inflammation damages epithelial cells through the production of reactive oxidative species (ROS), reactive nitrogen species (RNS), and proinflammatory cytokines. These products cause DNA damage and cell death, further destroying the epithelial barrier. *Pg* promotes the onset of chronic periodontal disease, which is characterized by a systemic antibody response, measured by the serum levels of antibodies against periodontal pathogens [7,8].

The immune system preserves the microenvironment for the microbiota maintaining local and systemic homeostasis and preserving host biological integrity [9]. The presence of serum antibodies to major periodontal pathogens has been associated with heart disease, stroke [10,11,12], and Alzheimer’s Disease (AD) [13,14]. In particular, elevated levels of immunoglobulin G (IgG) against *Pg* were detected in subjects before cognitive impairment [15,16]. Anti-*Pg* antibodies were also detected in neurological patients’ sera using enzyme-linked immunosorbent assays (ELISA) [17].

Specific oral bacterial species have also been implicated in several systemic diseases, such as bacterial endocarditis [18], aspiration pneumonia [19], osteomyelitis in children [20], pre-term low birth weight [21], obesity, diabetes [22], cardiovascular disease [23], rheumatoid arthritis, osteoporosis [24,25,26], and neurological and psychiatric disorders such as autism spectrum disorder, bipolar disorder, post-traumatic stress disorder, schizophrenia, and major depressive disorder [27,28,29]. In recent years, many researchers have introduced the oral microbiota–brain axis, given that the oral cavity is the origin of the gastrointestinal tract and could represent an extension of the microbiota gut–brain axis. In normal conditions, the oral and gut microbiome profiles are well-segregated due to the oral–gut barrier, physical distance, and chemical impediments, such as gastric acid and bile. In pathological conditions, oral–gut barrier impairment can allow for bidirectional communication between the oral and gut microbes, reshaping the microbial ecosystems of both habitats [30]. Thus, the mechanisms through which oral bacterial composition can impact the brain can be direct or indirect. The direct mechanism could be related to the trigeminal/olfactory/facial nervous system and the bloodstream [29]. The indirect mechanism could be related to the involvement of gut microbiota dysbiosis and systemic inflammation [31]. Both mechanisms could promote the release of proinflammatory cytokines, such as interleukin-IL-1β, IL-6, tumor necrosis factor (TNF)-α, chemokines, inflammasome NLRP3, and ROS, exacerbating neuroinflammation and provoking synaptic toxicity and neuronal death [29]. The microbiota–gut–brain axis and the oral microbiota–brain axis play important roles in maintaining homeostasis and their dysfunctions have been linked to various central nervous system (CNS) diseases, such as multiple sclerosis, AD, and Parkinson’s Disease (PD) [32,33,34,35]. It is well known that *Pg*-induced inflammation in the oral cavity can be transferred to the CNS. *Pg* lipopolysaccharide (LPS) can reach the brain-resident microglia via the leptomeningeal cells, which express *Pg*-LPS receptors (TLR2 and TLR4). Liu et al. demonstrated that these cells can transmit inflammatory signals from peripheral macrophages to the brain-resident microglia due to *Pg*-LPS stimulation [36]. Furthermore, more LPS from *Pg* has been detected in AD patients’ brain tissues compared to healthy volunteers [37]. AD incidence and mortality risk have been linked to a composite of *Pg* titers and neuroinflammation processes [38].

Recently, a novel pathogenic model, brain-first versus body-first, was proposed to classify two different subtypes of PD. According to this model, in the brain-first (top-down) type, α-synuclein pathology could initially arise in the brain with secondary spreading to the peripheral autonomic nervous system. In the body-first (bottom-up) type, the pathology could originate in the enteric or peripheral autonomic nervous system and then spread to the brain. However, these findings are still under further investigation [39].

Proactive approaches to balancing oral dysbiosis include the use of probiotics, paraprobiotics, postbiotics, and ozone therapy [40].

In this study, we evaluated the immune system’s response to *Pg* abundance in the oral cavities of patients affected by pathologies in the CNS. In particular, anti-*Pg* antibodies in the serum and the *Pg* abundance in the oral cavity were quantified in patients with neurodegenerative diseases compared to healthy controls (HC) and patients affected by acute or chronic neurological diseases. 

## 2. Materials and Methods

### 2.1. Study Cohort

The study was conducted according to the guidelines of the Declaration of Helsinki and approved by the local Ethics Committee of the “G. d’Annunzio” University of Chieti-Pescara on May 2020 (ethics code: 1909).

A previous pilot study [17] found a significant difference (*p* = 0.037, with a statistical power of 60.4% considering the probability of type I error *α* = 0.05) between 15 patients affected by neurodegenerative diseases and 21 patients affected by neurological non-neurodegenerative diseases in the ratio between anti-*Pg* antibodies’ quantity and *Pg* abundance. In the same condition, the statistical power could reach 80% with *α* = 0.05 if a sample size of 33 was considered for each group. In addition, also considering the HC group, we estimated that, with a low percentage increase of 2.4% in the ratio values, we could obtain a statistical power of 80% and *α* = 0.05 with a sample size of 30 for each group. Accordingly, the eligibility criteria were evaluated in 30 HC and 90 neurological patients, who could be subdivided into three groups, obtaining 1:1 matching among the groups. HCs free from neurological diseases were recruited from a list of HCs. In contrast, from May 2020 to July 2022, all patients were enrolled in the Neurology Clinic of “SS Annunziata” Hospital of Chieti. Individuals under antibiotic therapy (2 HCs and 8 patients) or those using daily chlorhexidine mouthwash within the last 3 months (3 HCs and 7 patients) were excluded from the study. In addition, 5 patients were excluded for concomitant tumors, and 2 HCs and 6 patients declined to participate. Thus, the final cohort included 23 HCs, 17 patients with an acute neurological condition (N-AC), 19 patients with a chronic neurological condition (N-CH), and 28 patients with a neurodegenerative disease (N-DEG). Written informed consent was obtained from all 87 participants. For each individual, tongue biofilm was used to quantify the *Pg* abundance and a blood sample was used to quantify the anti-*Pg* antibody levels.

The N-AC group included patients referred to the Neurology Clinic for the following reasons: ischemic stroke (*n* = 7), head trauma (*n* = 6), encephalitis (*n* = 2), loss of consciousness (*n* = 1), and cerebrospinal fluid hypotension (*n* = 1).

The N-CH group included patients with a diagnosis of a chronic neurological condition such as: epilepsy (*n* = 7), myasthenia (*n* = 6), polyneuropathy (*n* = 4), brain tumor (*n* = 1), and vascular leukoencephalopathy (*n* = 1).

The N-DEG group included patients with the following diagnoses: Parkinson’s disease (*n* = 10, of which *n* = 5 had dementia), multiple sclerosis (*n* = 6), sclerosis lateral amyotrophic (*n* = 4), mixed dementia (*n* = 3), Alzheimer’s disease (*n* = 2), Frontotemporal dementia (*n* = 2), and Huntington’s disease (*n* = 1). 

Figure 1 shows a flow diagram of the case–control study from the eligibility screening to the collection of samples and oral indices. STROBE Statement for observational studies was provided in Appendix A.

### 2.2. Collection of the Samples

Tongue biofilm was taken from each patient and each HC 8 h after they last brushed their teeth. The swab was collected by brushing from the middle third of the tongue dorsum 5 times. After shaking it vigorously for 30 s, the swab was immediately transferred into 5.0 mL of phosphate-buffered saline (PBS) and then kept at +4 °C until the nucleic acid extraction. A blood sample was also taken from all the participants on the same day of the brushing and stored at −80 °C.

### 2.3. Oral Examinations

Oral examinations were performed by a trained dentist (P.P.) using mirrors (MIR3HD, Hu-Friedy, Chicago, IL, USA), a dental probe (PCP-UNC 15, Hu-Friedy, Chicago, IL, USA), and an intra-oral light. The teeth number, plaque index (PI), and gingival index (GI) were recorded using the standardized Oral Health Questionnaire [41]. All other indices were evaluated as described in Franciotti et al., 2021 [17].

### 2.4. Bacterial DNA Quantification on Brushing

In this study, the reference bacterial strain *Pg* ATCC 33277 (LGC Standards S.r.l., Sesto San Giovanni, Milano, Italy) was used and cultivated as previously reported [42]. Briefly, the bacterium was cultivated in Fastidious Anaerobe Agar (Neogen, Lansing, MI, USA) supplemented with 5% of defibrinated horse sterile blood (Oxoid Limited, Hampshire, UK) for 48 h in anaerobiosis (Anaerogen Pak Jar, Oxoid, UK). After incubation, all the colonies were collected, washed in PBS, and centrifugated to obtain a pellet corresponding to 6 × 10^12^ CFU. Molecular analyses were performed to quantify the *Pg* in the oral cavities of all the participants. The total genomic DNA was isolated from the samples and the reference bacterial strain using a Quick DNA miniPrep Plus KIT (Zymo Research, Irvine, CA, USA). StepOne™ 2.0 (Applied Biosystems, Thermo Fisher Scientific, Waltham, MA, USA) was used in a qPCR analysis to quantify the *Pg* abundance in each sample. A TaqMan-based assay that recognizes *Pg* 16S rRNA, the gene encoding the small subunit of 16S ribosomal RNA, was used, as previously reported [43]. A standard curve passing through 5 points was constructed, indicating the cycle threshold values versus the *Pg* 16S rRNA gene. This method allowed us to estimate the bacterial quantity based on the amount of total DNA isolated from the oral samples [43].

### 2.5. Antibody Assay on Serum 

The ELISA technique assayed the serum IgG antibody responses to the bacterial pathogens from *Pg* using the ChonBlock^TM^ buffer system (Human Anti-Bacteria & Toxins Antibody ELISA Kits, #6119, Chondrex, Woodinville, WA, USA, an ELISA protocol to improve the accuracy and reliability of serological antibody assays) as follows [44]. Specific 96-well plates were used to perform the assay at room temperature. Dilution samples less than 1:1000 required a parallel run on both antigen-coated and uncoated plates; conversely, for dilution samples more than 1:1000, only antigen-coated plates were run. In detail, sera were collected from each participant and stored at −80 °C. The sera were then thawed and centrifuged at 10,000 rpm for 5 min to perform the assay. The supernatants were recovered and suitably diluted with ChonBlock^TM^ standard/sample dilution buffer. Moreover, Blocking Buffer was added to each well and incubated for 1 h; the plate was washed using Wash Buffer; the standards and samples were loaded and incubated for 2 h; the plate was washed using Wash Buffer; secondary antibody solution was loaded and incubated for 1 h; the plate was washed using Wash Buffer; TMB solution was added and incubated for 25 min; and finally, Stop Solution was added and the OD values at 450 nm were recorded within 5 min. The antibody levels were determined by comparing them to standard levels and were analyzed using a regression analysis. The antibody levels were expressed as EU (10^3^ units/mL).

### 2.6. Statistical Analyses

The data were compared among the groups using non-parametric Kruskal–Wallis tests. Levene’s test was used to test the homogeneity of the variance among the groups. A post hoc test was applied for pairwise comparisons. Spearman’s correlation tests were performed on all the participants to evaluate the associations between the variables. Spearman’s correlation tests were also performed separately for each group to evaluate the effect of age and sex on the *Pg* abundance and anti-*Pg* antibody quantity. The level of significance for all the statistical analyses was set at 0.05.

## 3. Results

### 3.1. Participants’ Characteristics 

The demographic and oral characteristics of the study cohort are shown in Table 1.

Age significantly differed among the groups (χ^2^(3) = 10.4, *p* = 0.02). The post hoc comparisons showed that only the N-AC group was significantly older than HCs (*p* = 0.02). Sex, smoker status (%), and former smoker status (%) were not significantly different among the groups (χ^2^(3) = 4.4, *p* = 0.22, χ^2^(3) = 1.3, *p* = 0.73, and χ^2^(3) = 4.6, *p* = 0.21, respectively). The teeth number significantly differed among the groups (χ^2^(3) = 13.9, *p* = 0.003). The post hoc comparisons showed that HC group had a higher teeth number than the N-AC (*p* = 0.008), N-CH (*p* = 0.04), and N-DEG (*p* = 0.02) groups. No significant differences among the groups were found in the number of fixed prostheses (*p* = 0.27) and removable dentures (*p* = 0.17). Due to the low number of participants who completed the dental visit, statistical comparisons among the groups were not performed on other oral indices.

### 3.2. Pg Bacteria and Antibody Quantification 

The means by group of the *Pg* abundance, anti-*Pg* antibodies’ quantity, and ratio between the anti-*Pg* antibodies’ quantity and *Pg* abundance are shown in Table 2. 

Spearman’s correlation showed a significant positive correlation between the abundance of *Pg* and the anti-*Pg* quantity (ρ = 0.61, *p* = 0.0001). A significant correlation was also found between the *Pg* abundance, age (ρ = 0.24, *p* = 0.029), and sex (ρ = 0.21, *p* = 0.047). These correlations are shown in Figure 2. The abundance of *Pg* in the oral cavity increased with age and in males. A significant correlation between age and *Pg* abundance was found for the HC group only (ρ = 0.47, *p* = 0.025). Indeed, the N-AC, N-CH, and N-DEG groups showed no significant correlations between *Pg* abundance and age (ρ = −0.13, *p* = 0.62; ρ = 0.34, *p* = 0.15; and ρ = −0.11, *p* = 0.59, respectively). No significant correlation was found between *Pg* abundance and sex at the group level (ρ = 0.11, *p* = 0.60 for HC; ρ = 0.24, *p* = 0.35 for N-AC; ρ = 0.28, *p* = 0.26 for N-CH; and ρ = 0.11, *p* = 0.59 for N-DEG). No significant correlation was found between the abundance of *Pg* and smoking (*p* = 0.35), prior smoking (*p* = 0.99), teeth number (*p* = 0.44), number of fixed prostheses (*p* = 0.54), and number of removable dentures (*p* = 0.22). No significant correlation was found between anti-*Pg* antibodies’ quantity and age (*p* = 0.90), sex (*p* = 0.07), smoking (*p* = 0.54), or former smoking (*p* = 0.53). 

Non-parametric Kruskal–Wallis tests were performed to compare the *Pg* bacteria and antibody quantities among the groups. Specifically, logarithmic transformations were applied to the *Pg* abundance value, anti-*Pg* antibodies’ quantity, and the ratio between anti-*Pg* and *Pg* to respect the homogeneity of the variance among the groups, as assessed by Levene’s test. The Kruskal–Wallis H test showed that the log abundance of *Pg* in the oral cavity was significantly different among the groups (χ^2^(3) = 31.9, *p* = 0.0001), with a mean rank value of 23.1 for the HC, 38.2 for the N-AC, 48.8 for the N-CH, and 61.4 for the N-DEG groups. The post hoc test showed that the *Pg* abundance in the oral cavity was higher in the N-DEG group than the HC (*p* = 0.0001) and N-AC (*p* = 0.01) groups, and it was also higher in the N-CH group than the HCs (*p* = 0.005).

The log of anti-*Pg* antibodies was significantly different among the groups (χ^2^(3) = 10.0, *p* = 0.020), with mean rank values of 31.8, 42.8, 55.9, and 46.7 for the HC, N-AC, N-CH, and N-DEG groups, respectively. The post hoc test showed that the anti-*Pg* antibodies in the serum were higher in the N-CH group than the HCs (*p* = 0.012).

The log of the ratio between the anti-*Pg* antibodies’ quantity and the *Pg* abundance was significantly different among the groups (χ^2^(3) = 37.5, *p* = 0.0001), with mean rank values of 65.4 for the HC, 50.4 for the N-AC, 43.8 for the N-CH, and 22.7 for the N-DEG groups. The post hoc test showed that the log of the ratio between the anti-*Pg* antibodies’ quantity and the *Pg* abundance was lower in the N-DEG group than the HC (*p* = 0.0001), N-AC (*p* = 0.002), and N-CH (*p* = 0.03) groups, and it was also lower in the N-CH group than the HCs (*p* = 0.04). 

Figure 3 shows the scatterplots, the medians, and the significant results of the *Pg* abundance logarithm, the anti-*Pg* antibodies’ logarithm, and the logarithm of the ratio between the anti-*Pg* antibodies and *Pg* abundance for each group. 

## 4. Discussion

This study showed that the *Pg* abundance in the oral cavity was higher in people with chronic neurological conditions than in the HCs or people affected by an acute neurological condition. Previous studies have shown that the oral microbiota composition originating from buccal and sublingual mucosa and salivary samples differs between healthy subjects and PD patients [45,46,47,48]. Saliva represents a collection of the microbiota from all the ecological niches in the oral cavity and is easy to collect. However, it has the major disadvantage of lacking specificity compared to samples collected from a single oral site. The use of brushes allows for a noninvasive, simple, and repeatable way of managing abundant site-specific microbiota [49,50]. Unlike the salivary microbiota, the supragingival dental microbiota are subject to more variation, probably due to brushing habits [51].

Periodontal pathogens, such as *Pg* and Treponema denticola, are increased in subjects with cognitive impairment or AD [52,53]. The action of accessory pathogens could facilitate *Pg* colonization, and commensal-turned pathobionts could contribute to inflammation [3]. It has been suggested that oral dysbiosis is unlikely to result from dental or periodontal complications in patients affected by cognitive impairment or AD. Instead, it could be a primary event related to the pathogenesis of neurodegenerative diseases [54]. The mechanism by which bacteria cross the blood–brain barrier remains unclear. A possible explanation may lie in gingipains, the class of *Pg* cysteine proteases. A *Pg* increase in neurological patients may lead to a higher delivery of gingipains, which are involved in the degradation of the tight junction proteins of cerebral microvascular endothelial cells (i.e., Zonula occludens-1 and occluding), promoting damage to the blood–brain barrier [55]. *Pg* produces two forms of gingipains: the free form and the outer membrane vesicles (OMVs)-associated form. Probably, via the OMVs, gingipains may be transported and released, increasing the permeability of human cerebral microvascular endothelial cells [42].

In our study, *Pg* abundance was also increased with age, confirming previous findings [56]. However, the analysis of single groups indicated that the correlation between *Pg* abundance and age was significant for the HCs only. This result suggests that neurological conditions bother the relationship between age and the abundance of *Pg* bacteria in the oral cavity. The sex effect could have been related to the greater number of males in the N-DEG group, which could have biased the result toward an increased abundance of *Pg* in males compared to females. Indeed, when the correlation analyses were performed on single groups, the correlation between *Pg* abundance and sex was lost.

A significant positive correlation was found between the abundance of *Pg* in the oral cavity and the quantity of anti-*Pg* antibodies in the serum of all the participants, confirming the hypothesis that oral bacteria can cause the production of anti-*Pg* antibodies, as previously reported [17,57]. In addition, this relationship confirms that the number of bacteria is also influenced by the immune system’s ability to manage the host’s state [58].

Interestingly, despite the higher amount of *Pg* in the N-DEG group, the quantity of anti-*Pg* antibodies was not higher in this group than the others. The inadequate response of the immune system of the N-DEG group in producing anti-*Pg* antibodies was revealed by the analysis of the ratio between the anti-*Pg* antibodies’ quantity and the *Pg* abundance. Thus, a low level of antibodies could result in a chronic inflammatory state inside and outside the oral cavity, which might be involved in the processes of neurodegeneration. Elevated IgG antibodies against periodontal disease bacteria have been found in subjects years before cognitive impairment, whereas IgG against *Pg* was decreased in patients who developed AD [15]. Chronic inflammation is an important factor in neurodegeneration, causing the dysregulation of circulating inflammatory molecules and the innate immune response [59,60]. Thereby, in turn, the LPS and gingipain of *Pg* can regulate immune responses by creating an inflammatory environment through the stimulation of TLR2-PI3K-mediated signaling, which results in a reduction in bactericidal activity, an increase in proinflammatory cytokines (e.g., IL-1β, IL-6, and TNF-α), and the inhibition of phagosome formation and maturation [61].

The N-CH group showed a higher quantity of anti-*Pg* antibodies than the HCs. However, the amount of these antibodies was inadequate compared to the HCs, as indicated by the significant difference between the N-CH and HC groups in the ratio between the anti-*Pg* antibodies and *Pg*. However, this ratio was higher in the N-CH than in the N-DEG group, suggesting that the low production of anti-*Pg* antibodies with a high abundance of *Pg* in the oral cavity was related more to neurodegeneration than to chronic neurological disease. On the contrary, the patients affected by an acute neurological condition (i.e., ischemic stroke or head trauma) included in the N-AC group showed no significant difference compared to the HC group. Indeed, the HC and N-AC groups showed higher antibodies in the presence of a low abundance of *Pg* in the oral cavity.

The accumulation of bacterial plaques by periodontopathogen bacteria, in addition to cigarette smoking, is the most important risk factor in causing an inflammatory response [62,63]. To reduce the bacterial load of the orange and red complex and to avoid consequent tissue damage, ozone-based therapies have been proposed in recent years and have been tested as antimicrobial therapy. Ozone produces free radicals that penetrate inside bacterial DNA and RNA, destroying their structure, thus accelerating curative action [64,65]. Ozone therapy as an adjunct to mechanical therapy reduces the gingival index in generalized chronic periodontitis patients [66].

Other therapeutic approaches include prebiotics, probiotics, paraprobiotics, and post-biotics. An example of probiotic therapy is *Lactobacillus reuteri*, which can reduce the amount of periodontopathogens and proinflammatory cytokines [55,67]. All these therapies exert an immunomodulatory and anti-inflammatory action on the periodontal tissues due to their antioxidant action [68,69,70].

The main limitation of our study is its relatively small sample size and the heterogeneity of the patient groups. Unfortunately, a considerable amount of participants were excluded or dropped out and the recruitment could not continue. Thus, we could not draw definitive conclusions about this oral bacterium’s influence and the altered immune response in the pathogenesis of neurodegenerative diseases. Future follow-up studies could monitor the microbiota profile and the immune response over time and at different stages of disease to investigate the influence of periodontopathogen bacteria on the onset and progression of neurological and neurodegenerative diseases. 

In conclusion, the response of the immune system to *Pg* abundance in the oral cavity, assessed by the quantification of anti-*Pg* antibodies in the serum, showed a stepwise model. The immune system response decreased progressively in patients affected by an acute condition such as ischemic stroke or head trauma, less in patients with epilepsy or peripheral nervous system disorders, and finally, even less in patients affected by neurodegenerative diseases such as Parkinson’s or multiple sclerosis. 

## Figures and Tables

**Figure 1 microorganisms-11-02555-f001:**
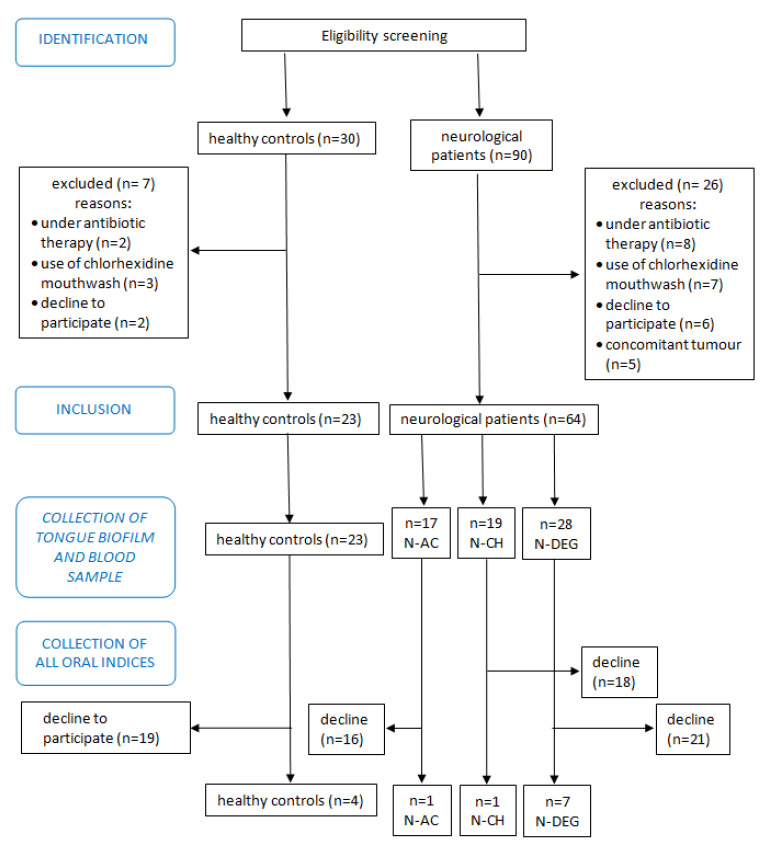
Flowchart of the study cohort.

**Figure 2 microorganisms-11-02555-f002:**
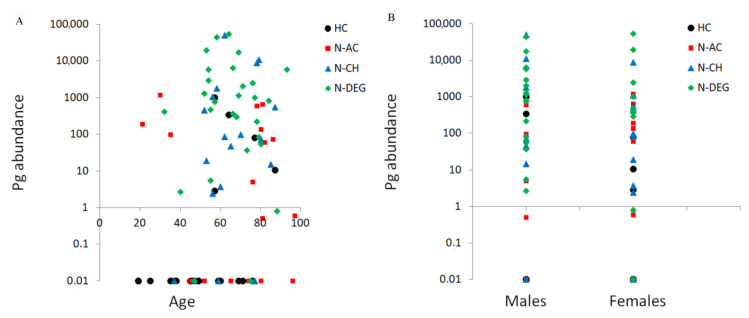
*Pg* abundance in logarithmic scale as a function of age (**A**) and sex (**B**).

**Figure 3 microorganisms-11-02555-f003:**
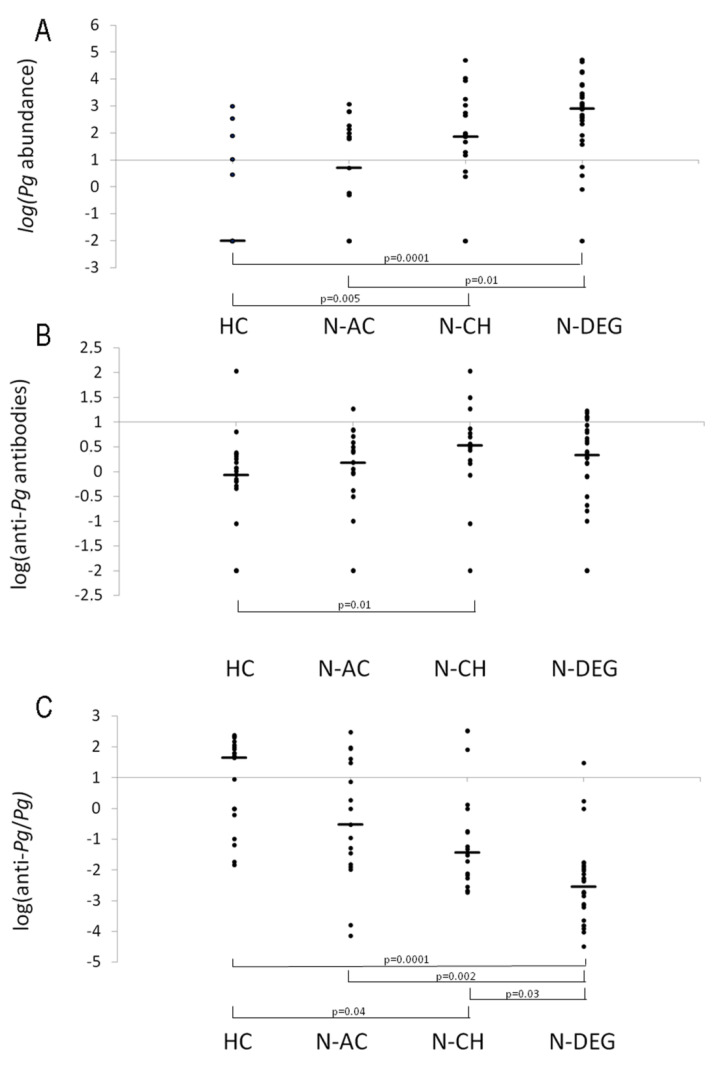
Scatterplot and significant results of the *Pg* abundance logarithm (**A**), the anti-*Pg* antibodies’ logarithm (**B**), and the logarithm of the ratio between anti-*Pg* antibodies and *Pg* abundance (**C**) for all the groups. Horizontal bars indicate the median value for each group.

**Table 1 microorganisms-11-02555-t001:** Demographic characteristics and oral indices of all participants of the groups included in the study. Otherwise, the number of patients included in the calculation is shown in parentheses.

	HC (*n* = 23)	N-AC (*n* = 17)	N-CH (*n* = 19)	N-DEG (*n* = 28)
Age	51.9 ± 3.8	68.2 ± 5.6	64.4 ± 3.1	65.5 ± 2.7
Sex (% male)	34.8	35.3	42.1	60.7
Smoker (%) Former smoker (%)	4.3 13.0	11.8 17.6	6.3 (*n* = 16) 31.3 (*n* = 16)	12.5 (*n* = 24) 17.5 (*n* = 24)
Teeth number	24.4 ± 1.8 (*n* = 20)	13.8 ± 3.0 (*n* = 16)	17.2 ± 2.2 (*n* = 15)	16.8 ± 1.9 (*n* = 26)
Plaque index	1.6 ± 0.4 (*n* = 5)	0.1 ± 0.0 (*n* = 1)	2.2 ± 0.2 (*n* = 7)	2.1 ± 0.3 (*n* = 8)
Gingival index	0.8 ± 0.6 (*n* = 5)	0.0 ± 0.0 (*n* = 1)	1.1 ± 0.2(*n* = 7)	1.1 ± 0.3(*n* = 7)
Presence of gingivitis (%)	60.0 (*n* = 5)	60.0 (*n* = 10)	75.0 (*n* = 12)	86.7 (*n* = 15)
Lingual patina index	1.2 ± 0.5 (*n* = 5)	0.8 ± 0.2 (*n* = 10)	0.8 ± 0.3 (*n* = 12)	1.9 ± 0.3 (*n* = 15)
Presence of oral infection (%)	60.0 (*n* = 5)	40.0 (*n* = 10)	66.7 (*n* = 11)	80.0 (*n* = 14)
Oral hygiene index	40.0 (*n* = 5)	77.8 (*n* = 9)	36.4 (*n* = 11)	14.3 (*n* = 14)
Presence of fissured tongue (%)	0.0 (*n* = 5)	20.0 (*n* = 10)	18.2 (*n* = 11)	0.0 (*n* = 15)
Number of fixed prostheses	0.2 ± 0.1 (*n* = 20)	0.3 ± 0.1 (*n* = 16)	0.6 ± 0.2 (*n* = 14)	0.4 ± 0.1 (*n* = 25)
Number of removable dentures	0.2 ± 0.1 (*n* = 19)	0.7 ± 0.3 (*n* = 14)	0.3 ± 0.2 (*n* = 14)	0.3 ± 0.1 (*n* = 24)

Values are expressed as mean ± standard error.

**Table 2 microorganisms-11-02555-t002:** *Pg* abundance and anti-*Pg* antibodies’ quantity of all participants of the groups included in the study.

	HC (*n* = 23)	N-AC (*n* = 17)	N-CH (*n* = 19)	N-DEG (*n* = 28)
*Pg* abundance (CFU/mL)	63.4 ± 46.1	178.4 ± 80.9	3961.4 ± 2698.8	6013.3 ± 2488.3
Anti-*Pg* antibodies (units/mL)	5.8 ± 4.6	3.2 ± 1.1	10.9 ± 5.6	4.7 ± 1.0
Anti-*Pg*/*Pg* (units/CFU)	69.5 ± 17.8	34.2 ± 18.9	40.1 ± 24.4	1.2 ± 1.1

Values are expressed as mean ± standard error.

## Data Availability

The data supporting this study’s findings are available upon reasonable request.

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
