# Peer review of "The Immune System Response to Porphyromonas gingivalis in Neurological Diseases"

_microorganisms, 2023, doi:10.3390/microorganisms11102555_

Round 1

Reviewer 1 Report (Previous Reviewer 2)

Thank you for sending me the resubmitted manuscript again, I had already given the ok for publication after the first round of review.

I remain available

Author Response

Thank you.

Reviewer 2 Report (New Reviewer)

Dear authors 

the article should be ammended strictly following the STROBE guidelines the checklist should be attached to the main text

https://www.strobe-statement.org/checklists/

The text is not organized, introduction is too long, conclusions are not supported by results and there is a long section about prebiotics that seems not justified since patients were not treated using such prebiotics.

The sample size is not justified by statistical means Moderate editing of English language required

Author Response

Reviewer 2

Dear authors

the article should be ammended strictly following the STROBE guidelines the checklist should be attached to the main text

https://www.strobe-statement.org/checklists/

In the revised version, we included the completed checklist of the STROBE guideline at the end of the main text.

The text is not organized, introduction is too long, conclusions are not supported by results and there is a long section about prebiotics that seems not justified since patients were not treated using such prebiotics.

We accommodated the Reviewer's request and streamlined and reorganized the introduction. We also modified the discussion/conclusions and shortened the section on prebiotics.

The sample size is not justified by statistical means

In the revised version, we justified the sample size statistically:

“A previous pilot study [29] found a significant difference (p=0.037, with a statistical power of 60.4%) between 15 patients affected by neurodegenerative diseases and 22 patients affected by neurological non-neurodegenerative diseases, on the ratio between anti-Pg antibodies quantity and Pg abundance. Thus, we estimated that with a sample size of 30, we could obtain a statistical power of 76% in comparing two patient groups with the same characteristics and higher statistical power in case-control comparison. Accordingly, we assessed for eligibility to the study 30 HC and 90 neurological patients, which could be subdivided into three groups, obtaining 1:1 matching among groups."

Comments on the Quality of English Language

English revisions were done by a colleague fluent in English writing.

This manuscript is a resubmission of an earlier submission. The following is a list of the peer review reports and author responses from that submission.

Round 1

Reviewer 1 Report

This submission fits into an interesting research topic. The text is well written, the references topical and well documented. One criticism, however, concerns the length of the introduction to the detriment of the discussion, whose references are few and far between. The two sections should be rebalanced.

The protocol poses a major scientific problem. This is an observational study which must comply with CONSORT criteria. Has the protocol been published or submitted to ClinicalTrials.gov?

Too much error when we know the sensitivity of the oral microbiota and more particularly of the tongue biofilm to clinical, biological, age, sex, oral hygiene, medication, gingival bleeding, CAL, depth of periodontal pockets, etc..... variables.

Focusing on Pg without considering the dependent variables published to date is a methodological error. 

And this without addressing the basic principles of this type of research, such as defining the population, the sample (chart flow), sample size calculation, inclusion and exclusion criteria, etc.

Reviewer 2 Report

  Manuscript of considerable interest for the dental sector, before proceeding with the evaluation of a possible publication, it needs a major revision. Abstracts; highlight the data more clearly Keywords: few, add specific ones that are registered on MeSH. Introduction: ok the substantial difference between gram positive and gram negative, it seems very basic to me, but substantially missing how these bacteria increase the progression of periodontal and pari-implant disease based on the new classification of 2017 and 2023, and add all the minimally invasive systems for professional and home treatment (Scribante et al.) Materials and methods: poorly described. Very confusing results: reorganize the tables and graphs (which are not in high resolution) so that they can be read by all readers, highlighting the data more clearly. Discussion: add as future goals, to reduce, but not completely eradicate, the use of probiotics, para-probiotics, postbiotics to reduce the incidence of the bacterial load of the orange and red complex, in order to reduce the progression of the disease periodontal and peri-implant as studied by Butera A. et al. Conclusions: add proactive action. Bibliography: add references required